# Pre-Sowing Laser Light Stimulation Increases Yield and Protein and Crude Fat Contents in Soybean

**Agnieszka Klimek-Kopyra** [1],*, **Reinhard W. Neugschwandtner** [2], **Anna Ślizowska** [1], **Dominika Kot** [1], **Jan Wincenty Dobrowolski** [3], **Zbigniew Pilch** [4] **and Ewa Dacewicz** [5]

[1] Department of Agroecology and Plant Production, University of Agriculture, Al. Mickiewicza 21, 31-120 Kraków, Poland

[2] Department of Crop Sciences, Institute of Agronomy, University of Natural Resources and Life Sciences Vienna (BOKU), 1180 Tulln, Austria

[3] Department of Photogrammetry Remote Sensing of Environment and Spatial Engineering, AGH University of Science and Technology, Al. Mickiewicza 30, 30-059 Kraków, Poland

[4] Faculty of Electrical and Computer Engineering, University of Technology, ul. Warszawska 24, 31-155, Kraków, Poland

[5] Department of Sanitary Engineering and Water Economy, University of Agriculture in Krakow, Al. Mickiewicza 24/28, 30-059 Krakow, Poland

* Correspondence: agnieszka.klimek@urk.edu.pl

**Abstract:** Pre-sowing laser light stimulation is a method commonly used to increase the productivity of legume species. However, it has not been proved that single-laser light stimulation is a more effective technique for enhancing plant productivity and seed yield quality than using different laser lights in sequence, by which means synergistic relations are produced. A two-year, single-factor field experiment was performed in order to test selected combinations of light stimulation of soybean seeds, the effectiveness of which would be expressed as increased plant yield and improved quality parameters. Pre-sowing light stimulation of soybean seeds was shown to significantly influence the morphological traits of the plants. It significantly increased pod number, pod weight, seed number, and seed weight compared to the control. Coherent laser light stimulation of soybean seeds with a helium–neon laser and with an argon laser increased soybean yields by 30% on average compared to the control. The ratio of the yield from the main shoot to the yield from the lateral branches in this treatment was 40:60, indicating that pre-sowing stimulation of seeds had a significant and positive effect by increasing the yield of the lateral branches relative to the control. Stimulation of seeds ($3 \times 3$ s) with a helium–neon laser significantly increased protein content in soybean seeds, on average by 11% compared to the control. A longer duration of pre-sowing stimulation of seeds ($3 \times 9$ s) resulted in a significant increase in crude fat content in the seeds by an average of 5% compared to the control. The use of physical light stimulation on soybean seeds is a promising solution for increasing soybean yields.

**Keywords:** soybean; light stimulation; yield components; shoots

## 1. Introduction

In recent years, there has been an increased interest in new technologies in agriculture aimed at improving crop yields while minimizing negative environmental impacts. The use of biotechnological methods in the cultivation of cereal, protein, and oilseed crops has numerous advantages, which unquestionably include an increase in the adaptability of plants to cultivation under various natural stresses, such as drought, temperature extremes, and mineral nutrient deficiency [1,2]. Soybean, as a species from Asia, has specific temperature and moisture requirements that determine its productivity. The short-term presence of stress factors during seed germination (drought, excessive rainfall, or cold snap) adversely affects the initial development of seedlings, while long-term effects of

certain stress factors reduce yields [3–6]. These phenomena occur in Central Europe and are the main limiting factors in the cultivation of soybean [4].

Various seed-conditioning methods are used to enhance the germination potential of soybean seeds, including biological and physical methods. One method, which has gained in importance in recent years, is the stimulation of seeds using low-power lasers emitting red, blue, or green light [7]. The laser light causes seeds to absorb light energy and convert it to chemical energy. The seeds use the additional energy supplied during irradiation in the germination process and in plant growth and development, resulting in a positive influence on these parameters [8,9]. The effect of stimulation with laser light depends on the parameters (laser intensity, wavelength, beam type, power density) and the method (He-Ne laser, diode laser) used [7].

Studies involving the irradiation of seeds for sowing usually use helium–neon (He-Ne) lasers with red light (with a wavelength of $\lambda = 632$ nm) or diode lasers with blue light (with a wavelength of $\lambda = 437$ nm). Beneficial effects on the metabolic processes of seeds and on the growth and development of plants, as well as increases in percentages of germinated seeds, have been noted following the use of He-Ne lasers [10–14].

Most of the studies reported in the literature have investigated the role of physical light stimulation in the improvement of seeds for sowing (i.e., increasing seed emergence), mainly under strictly controlled conditions [7]. This is probably due to the fact that the research results generated in laboratory conditions are repeatable, whereas, under field conditions, results can be erroneous due to variability in soil and weather conditions [15–17]. Laboratory analyses mainly determine the effects of the irradiation of seeds on the rate and uniformity of germination and on improvements to the biometric traits of seedlings. Field experiments, on the other hand, make it possible to determine the effects of the irradiation of seeds on additional parameters. These may include the speed of germination, flowering and maturation of plants, the length of the growing period, resistance to disease and pests, yield, and effects on parameters of yield structure [18–20].

The literature contains many studies that have investigated the effect of laser light on the morphological parameters of legume plants, such as white lupine or broad bean [21–25]. Podleśny and Stochmal [21] and Podleśny et al. [25] showed that irradiation of faba bean and white lupine seeds had a positive influence on the rate and uniformity of germination and the lengths and dry weights of the roots and the aerial parts of the seedlings [21,23]. It has been proved that pre-sowing irradiation considerably increased the activities of amylolytic enzymes and indole-3-acetic acid (IAA) contents in both species of seeds. Asghar et al. [5] revealed the pre-sowing seed-treatment effects on soybean sugar, protein, nitrogen, hydrogen peroxide ascorbic acid (AsA), proline, phenolic, malondialdehyde (MDA), and chlorophyll contents (Chl a and b). The authors highlighted that laser pre-sowing seed treatments have the potential to enhance soybean biological moieties and metabolically important enzymes; however, further studies should be undertaken to focus on the growth characteristics and yield attributes of soybean species.

In the available scientific literature, there are few reports confirming the positive effects of pre-sowing light stimulation with different time intervals and a combination of different types of lasers on the size and quality of soybean yield, depending on the location of shoots (main shoot vs. side shoots) on the plant. Therefore, the aim of the present study was to test the selected light stimulation treatments in soybean seeds, the effectiveness of which would be expressed as an increase in the productivity of the plants and improvements in their qualitative parameters.

## 2. Material and Methods

### 2.1. Experimental Design

A two-year (2018–2019), single-factor field experiment (Experimental Station of the Agricultural University in Prusy, northern Krakow, located in Southern Poland (47°24 N Lat., 7°19 E Long., 300 m a.s.l.)) was set up according to the randomized block method, with 4 replications. The experimental factor was the type of light stimulation. Six different

combinations (S0-S5) with different light beams and irradiation times were used: (1) the control, with no irradiation; (2) red laser light (LR) applied three times, 3 s each, with 3 s intervals between laser exposures (LR 3 × 3); (3) red laser light (LR) applied three times, 9 s each, with 9 s intervals between exposures (LR 3 × 9); (4) red laser light (LR) applied three times, 30 s each, with 30 s intervals between exposures (LR 3 × 30); (5) red laser light (LR) applied three times, 3 s each, with 3 s intervals between exposures (LR 3 × 3), followed by blue laser light (LB) applied three times, 1 s each, with 1 s intervals between exposures (LD 3 × 1); (6) blue laser light (LD) applied three times, 1 s each, with 1 s intervals between exposures (LD 3 × 1), followed by red laser light (LR) applied three times, 3 s each, with 3 s intervals between exposures (LR 3 × 3) (Table 1). The seeds were exposed to two types of lasers, according to Dłużniewska et al. [26], using the method presented by Klimek-Kopyra et al. [7]: a helium–neon laser (He-Ne, referred to as LR) producing red light with a wavelength of 632.8 nm and a density of irradiation of 2 W m$^{-2}$ and an argon laser (Ar, referred to as LD) producing blue light with a wavelength of 514 nm and a density of irradiation of 5 W m$^{-2}$.

**Table 1.** Experimental design.

| Treatment of Seed Light Stimulation Before Sowing (S) | Laser Type | Laser Exposure Time of Light Stimulation |
|---|---|---|
| **S0 Control** | - | - |
| **S1 (LR 3 × 3)** | He-Ne | 3 × 3 s * |
| **S2 (LR 3 × 9)** | He-Ne | 3 × 9 s |
| **S3 (LR 3 × 30)** | He-Ne | 3 × 30 s |
| **S4 (LR 3 × 3 + LD 3 × 1)** | He-Ne + Argon | 3 × 3 s + 3 × 1 s |
| **S5 (LD 3 × 1 + LR 3 × 3)** | Argon + He-Ne | 3 × 1 s + 3 × 3 s |

* Time units presented in SI units. The intervals between laser exposures were the same as the exposure times.

The field experiment was set up on chernozem soil characterized by high contents of plant-available phosphorus, potassium, and magnesium. The soil pH was 6.4 (pH in 1 mol dm$^{-3}$ KCl), and the nutrients in the topsoil layer were (per 100 g soil): phosphorus, 21.2 mg; potassium, 23.7 mg; and magnesium, 10.4 mg. The total mineral nitrogen content in the soil at a 0–90 cm depth was 74.7–78.9 kg ha$^{-1}$. Due to the optimal conditions for plant growth and development, no mineral fertilizers were used.

Before sowing (on the 5th of May 2018 and on the 8th of May 2019), uniform soybean seeds of the cultivar Augusta were subjected to physical light stimulation with low-power lasers in the laboratory: a helium–neon (He-Ne) laser (2 mW/m$^2$) with red light (LR) (632.8 nm) and an argon laser (5 mW/m$^2$) with blue light (LD) (514 nm). The beam diameter of each laser was the same: 5 mm. The area of each microplot was 7 m$^2$. The width between rows was 25 cm. The seed sowing density was 70 seeds per square meter; after germination, the final plant density was the same for each plot and equal to 50 plants per square meter. The irradiated seeds emerged at the same time—two weeks after sowing—while the seeds for the control treatment emerged five days later.

After the plants reached full maturity (on the 7th of September in 2018 and on the 10th of September in 2019), they were pulled out of the field manually (10 plants per plot) for the determination of biometric traits. The following biometric parameters were determined manually using lab scales and a digital ruler: root crown diameter, plant height, height of first pod setting, plant dry matter, and number of lateral branches. In addition, the biometric traits of shoots were determined, depending on their placement on the plant (main shoot versus lateral branches): dry weight of shoots, dry weight of stems, height of shoots, and number of fruiting nodes per shoot. Parameters for yield structure were also assessed, depending on the hierarchy of shoots (main versus lateral): number of pods, weight of pods, number of seeds, and weight of seeds. The yield per plant and the quality of the seeds (crude protein and crude fat content) were estimated [27]. The chemical

composition of the seeds was determined by near-infrared spectroscopy (NIRS) with an InfraXact$^{TM}$ analyzer (Foss$^®$) (Nils Foss Alle 1 DK-3400, Hilleroed, Denmark).

Statistical analysis of the results was performed using ANOVA. The significance of differences was determined by Tukey's multiple range test, using a significance level of $p = 0.05$. Average values of both years are presented (2018–2019). The obtained data were subjected to statistical analysis using the Statistica$^®$ package (a data analysis software system; TIBCO software Inc., Palo Alto, CA, USA), version 13.1.

Principal component analysis (PCA) was used to determine the relationships between the analyzed parameters. PCA is a method deployed to analyze a dataset containing both quantitative and qualitative variables. The results of the PCA included scores for the experimental plots and loading vectors of yield and yield components for the first and second principal components (PC1 and PC2), which are displayed in biplots. The loading vectors quantify the effects of PCs on yield and yield components. They also provide information on correlations between yield and yield components depending on the parameters of location on main shoots or lateral branches. Coinciding loading vectors indicate a positive correlation, while opposing loading vectors show a negative correlation. PCA was used to analyze the similarities between individuals by taking into account mixed-variable types. Statistica version 13.3 TIBCO software was used for the FA PCA analysis.

### 2.2. Environmental Conditions

Rainfall and temperature were analyzed for the growing period and years of the study (Figure 1). In the period analyzed (April–September), the precipitation totals showed little deviation from the long-term average of 445 mm (20 mm for 2019 and 30 mm for 2018). In 2018, the months of May, June, August, and September were assessed as average. Only April was extremely dry, while July was very wet. In 2019, the beginning of the growing period was very wet, but soil drought was observed in June, while July and August were average. The air temperature in 2018 deviated significantly from the long-term average—by +5.7 °C in April, +3.8 °C in May, +2.5 °C in June, +1.9 °C in July, +3.2 °C in August, and +2.3 °C in September. In 2019, the greatest deviations from the long-term average were observed in June (−5.9 °C) and August (−2.9 °C). In 2018, nearly the entire growing period was extremely warm, except for July, which was very warm. August was warm in both years. The year 2019 was more variable than 2018. The beginning, middle, and end of the growing period in 2019 (April, July, and September) were warm. The results of the research are presented as averages over the years due to the lack of unequivocal differentiation of the weather conditions in the prevailing growing seasons. No significant interaction between treatments and weather was detected. Thus, the results are presented as averages.

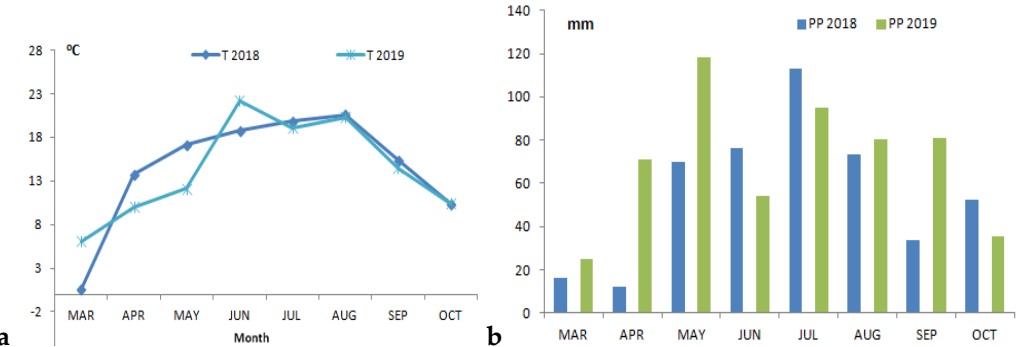

**Figure 1.** Average monthly temperature (**a**) and amount of precipitation (**b**) in the growing seasons 2018–2019.

### 3. Results

#### 3.1. Yield Depending on the Type of Light Stimulation

Pre-sowing light stimulation of soybean seeds with laser light had a positive effect on crop yields (Table 2). Both single stimulation with laser light and sequential laser

stimulation using red and blue lasers in various combinations were shown to be effective techniques for increasing plant productivity and seed yield quality.

**Table 2.** Yields of soybean plants.

| Treatment | Total Yield * (g m$^{-2}$) | Yield (g m$^{-2}$) | | TSW (g) | |
|---|---|---|---|---|---|
| | | Main Shoot | Lateral Branches | Main Shoot | Lateral Branches |
| S0 Control | 213a | 133a | 80a | 129a | 81c |
| S1 (LR 3 × 3) | 423ab | 199a | 223ab | 150ab | 129ab |
| S2 (LR 3 × 9) | 291a | 157a | 134a | 142a | 127abc |
| S3 (LR 3 × 30) | 648b | 270a | 378bc | 177b | 159b |
| S4 (LR 3 × 3 + LD 3 × 1) | 249a | 148a | 101a | 131a | 108ac |
| S5 (LD 3 × 1 + LR 3 × 3) | 689b | 268a | 420c | 180b | 144ab |

* Total yield is a sum of yield obtained from main and lateral branches of plant. Means in columns with different letters are significantly different ($p < 0.05$).

The use of combinations S3 (LR 3 × 30) and S5 (LD 3 × 1 + LR 3 × 3) led to higher plant yields. The use of combination S5 (LD 3 × 1 + LR 3 × 3) significantly increased the total yield, the yield from the lateral branches, and 1000-seed weight (TSW) from the main shoots. The use of combination S3 (LR 3 × 30) had a beneficial effect on yield from the main shoots and TSW from the lateral branches. Yields from the main shoots increased by 102% on average and TSW by 94%.

*3.2. Plant Biology*

Light stimulation of soybean seeds had a significant effect on the biometric features of the plants (Table 3). Significant increases in the weights and heights of the plants were observed for the selected light stimulation treatments, but the height of the first pod and the number of lateral branches was not affected by light stimulation. The use of treatment S2 (LR 3 × 9) significantly increased plant height, with an increase of 22.5 cm compared to the control. Although the heights of the first pods were not significantly affected by biostimulation, the combination S4 (LR 3 × 3 + LD 3 × 1) caused the first pod setting to be increased by 2 cm relative to the control. A positive effect of light stimulation on plant weight was observed following S3 (LR 3 × 30) and S5 (LD 3 × 1 + LR 3 × 3). The use of treatment S3 (LR 3 × 30) or S5 (LD 3 × 1 + LR 3 × 3) increased the number of lateral branches by 77% in comparison with the control (not significant).

**Table 3.** Biometric features of the plants depending on the type of light stimulation of seeds.

| Treatment | Plant Height (cm) | Height of First Pod (cm) | Dry Weight of Plant (g) | Number of Lateral Branches |
|---|---|---|---|---|
| S0 Control | 69.8a | 6.57a | 62.4a | 4.43a |
| S1 (LR 3 × 3) | 83.4ab | 7.86a | 119.9ab | 7.29a |
| S2 (LR 3 × 9) | 92.4b | 5.21a | 86.2a | 5.29a |
| S3 (LR 3 × 30) | 83.3ab | 5.57a | 162.6b | 7.86a |
| S4 (LR 3 × 3 + LD 3 × 1) | 84.1ab | 8.36a | 75.2a | 7.00a |
| S5 (LD 3 × 1 + LR 3 × 3) | 82.4ab | 6.79a | 176.1b | 7.86a |

Means in columns with different letters are significantly different ($p < 0.05$).

Physical light stimulation of seeds did not affect the biometric traits of the main shoots of soybean plants, which proves the high genotypic stability of the species (Table 4). In contrast, statistical analysis of selected biometric traits of the lateral branches showed that they were significantly affected by the experimental factor, which confirmed the high plasticity resulting from their high susceptibility to the physical factors and the lack of genotypic stability with regard to this feature (Table 5). Light stimulation of seeds resulted in a significant increase in the weight of the lateral branches, their size, and the number

of fruiting nodes. The most effective combination was S5 (LD 3 × 1 + LR 3 × 3). This treatment significantly increased the sizes of the lateral branches, including their heights and weights and the numbers of fruiting nodes.

**Table 4.** Selected biometric traits of the main shoots of plants grown from control and laser-stimulated seeds.

| Treatment | Shoot Weight (g) | Stem Weight (g) | Height (cm) | Number of Fruiting Nodes |
|---|---|---|---|---|
| S0 Control | 40.3a | 9.1a | 69.9a | 13.3a |
| S1 (LR 3 × 3) | 59.0a | 14.9a | 83.1a | 13.9a |
| S2 (LR 3 × 9) | 49.8a | 12.7a | 92.4a | 14.7a |
| S3 (LR 3 × 30) | 72.1a | 15.0a | 83.9a | 15.4a |
| S4 (LR 3 × 3 + LD 3 × 1) | 45.3a | 10.7a | 84.1a | 14.7a |
| S5 (LD 3 × 1 + LR 3 × 3) | 70.1a | 12.9a | 72.8a | 15.7a |

Means in columns with different letters are significantly different ($p < 0.05$).

**Table 5.** Selected biometric traits of the lateral branches of plants grown from control and laser-stimulated seeds.

| Treatment | Shoot Weight (g) | Stem Weight (g) | Height (cm) | Number of Fruiting Nodes |
|---|---|---|---|---|
| S0 Control | 4.63a | 0.59a | 21.2c | 3.46a |
| S1 (LR 3 × 3) | 9.40ab | 1.71bcd | 37.9ab | 6.46bc |
| S2 (LR 3 × 9) | 6.69a | 1.24abc | 35.2a | 5.11ab |
| S3 (LR 3 × 30) | 14.1bc | 2.09cd | 47.7b | 7.49cd |
| S4 (LR 3 × 3 + LD 3 × 1) | 4.20a | 0.80ab | 31.9ac | 4.48 a |
| S5 (LD 3 × 1 + LR 3 × 3) | 18.0c | 2.39d | 47.1b | 9.43d |

Means in columns with different letters are significantly different ($p < 0.05$).

Light stimulation of soybean seeds had a significant effect on yield structure, depending on the location of the shoot on the plant—main shoot versus lateral branches (Tables 6 and 7). Irrespective of the irradiation combination and the position of the shoots on the plant, pod number, pod weight, seed number, and seed weight were generally high in the control group.

**Table 6.** Elements of yield structure for the main shoots of plants grown from control and laser-stimulated seeds.

| Treatment | Pod Number | Pod Weight (g) | Seed Number | Seed Weight (g) |
|---|---|---|---|---|
| S0 Control | 4.13a | 2.47a | 10.7a | 1.53a |
| S1 (LR 3 × 3) | 5.67bc | 3.59bc | 14.7bc | 2.35b |
| S2 (LR 3 × 9) | 4.66ab | 2.74ab | 11.6ab | 1.73ab |
| S3 (LR 3 × 30) | 6.33c | 4.56c | 16.3c | 3.12c |
| S4 (LR 3 × 3 + LD 3 × 1) | 4.73ab | 2.57ab | 11.5ab | 1.61a |
| S5 (LD 3 × 1 + LR 3 × 3) | 6.11c | 4.58c | 16.1c | 3.16c |

Means in columns with different letters are significantly different ($p < 0.05$).

**Table 7.** Elements of yield structure for lateral branches of plants grown from control and laser-stimulated seeds.

| Treatment | Pod Number | Pod Weight (g) | Seed Number | Seed Weight (g) |
|---|---|---|---|---|
| S0 Control | 1.94a | 0.95ab | 4.57a | 0.58ab |
| S1 (LR 3 × 3) | 2.01ab | 1.02a | 4.91a | 0.66a |
| S2 (LR 3 × 9) | 2.03ab | 1.04a | 4.89a | 0.65a |
| S3 (LR 3 × 30) | 2.40bc | 1.51c | 6.31b | 1.04c |
| S4 (LR 3 × 3 + LD 3 × 1) | 1.74a | 0.73b | 3.94a | 0.45a |
| S5 (LD 3 × 1 + LR 3 × 3) | 2.44c | 1.34c | 6.00b | 0.90c |

Means in columns with different letters are significantly different ($p < 0.05$).

On main shoots (Table 5), a significant positive effect on the parameters of yield structure was observed following the use of treatments S3 (LR 3 × 30) and S5 (LD 3 × 1 + LR 3 × 3). These combinations, in comparison with the control, caused increases in the numbers of pods of 53% and 47%, and in the numbers of seeds of 52% and 50%, and in the weights of pods of 55% and 54%. The weights of the pods following stimulation with these combinations increased by 85%, while the seed weights increased by 103% and 106%.

On lateral branches (Table 7), the most beneficial effects on production were noted following the use of combinations S3 (LR 3 × 30) and S5 (LD 3 × 1 + LR 3 × 3). The plants had 23.7% and 25.7% more pods than the plants in the control group. The numbers of seeds increased by 38.1% and 31.3%, respectively. Light stimulation with combinations S3 (LR 3 × 30) and S5 (LD 3 × 1 + LR 3 × 3) was shown to have a very positive effect on seed weight. Combination S4 (LR 3 × 3 + LD 3 × 1) resulted in decreased numbers of pods, weights of pods, numbers of seeds, and weights of seeds (not significant) (Table 7). In comparison with the control, pod weights and seed weights decreased by 22.4%, seed numbers by 13.8%, and pod numbers by 10.3% (Table 6).

### 3.3. Chemical Composition of Seeds

Light stimulation influenced the chemical composition of the seeds obtained from the main shoots and the lateral branches (Figure 2). In the case of the main shoots, the use of combinations S1 (LR 3 × 3) and S5 (LD 3 × 1 + LR 3 × 3) significantly increased the crude protein contents of the soybean seeds, while S2 (LR 3 × 9) significantly increased the crude fat contents of the seeds in comparison to the control. Extending the irradiation time in treatment S3 (LR 3 × 30) increased the crude protein contents of the seeds by 14.4% relative to the control. On the other hand, the use of combination S4 (LR 3 × 3 + LD 3 × 1) had no effect on the crude protein or crude fat contents of the seeds.

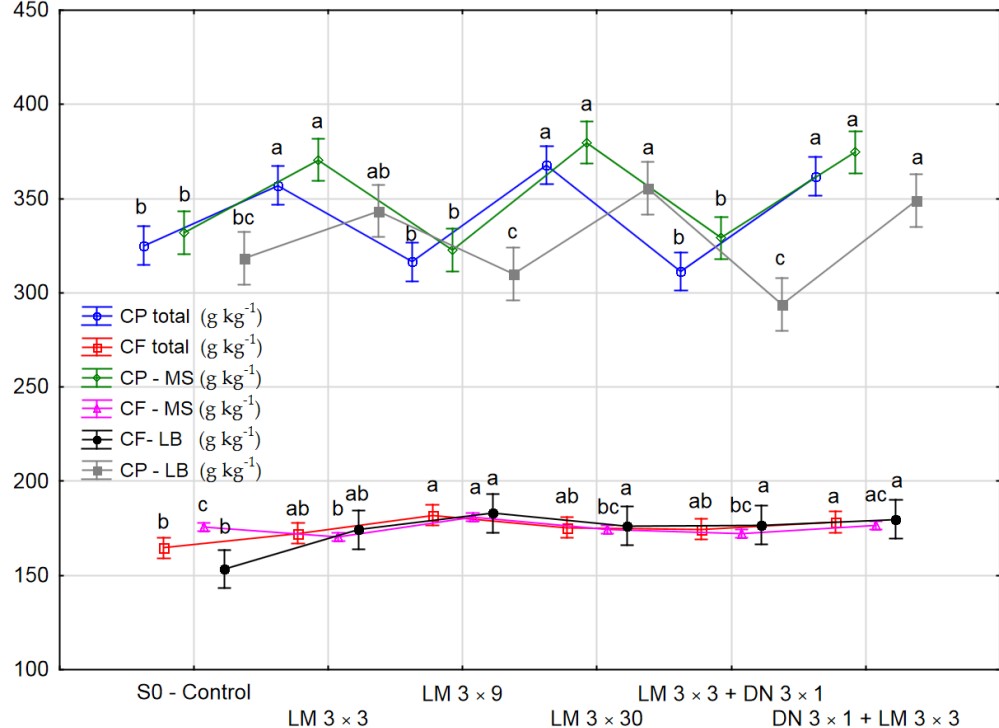

**Figure 2.** Selected quality traits of grain obtained from shoots (main versus lateral), depending on the type of seed stimulation. Crude protein (g kg$^{-1}$), average from plant (CP-Total); crude fat (g kg$^{-1}$), average from plant (CF-Total); crude protein (g kg$^{-1}$), main shoot (CP-MS); crude fat (g kg$^{-1}$), main shoot (CP-MS); crude fat (g kg$^{-1}$), lateral branches (CF-LB); crude protein (g kg$^{-1}$), lateral branches (CP-LB). Means in columns with different letters are significantly different ($p < 0.05$).

In the lateral branches, the crude protein and crude fat contents of the seeds were non-significantly lower than in the main shoots (Figure 2). The greatest increase in crude protein content, amounting to 11%, was obtained following the use of S3 (LR 3 × 30). A significant increase in crude fat content in the soybean seeds, amounting to 19%, was observed following the use of combination S2 (LR 3 × 9). A significant decrease in crude fat content was observed for the treatments in which combinations S2 (LR 3 × 9) and S4 (LR 3 × 3 + LD 3 × 1) were used.

### 3.4. PCA Analysis

The Principal Component Analysis (PCA) indicated different effects of laser stimulation on biological parameters (Figure 3). The obtained correlations between variables were extracted by default and are presented in Figure 3. The PCA analysis was conducted in six groups for the main experimental factor (irradiated material by laser) and presented the relationships between total yield and morphological parameters located on the main shoots (MSs) and lateral branches (LBs).

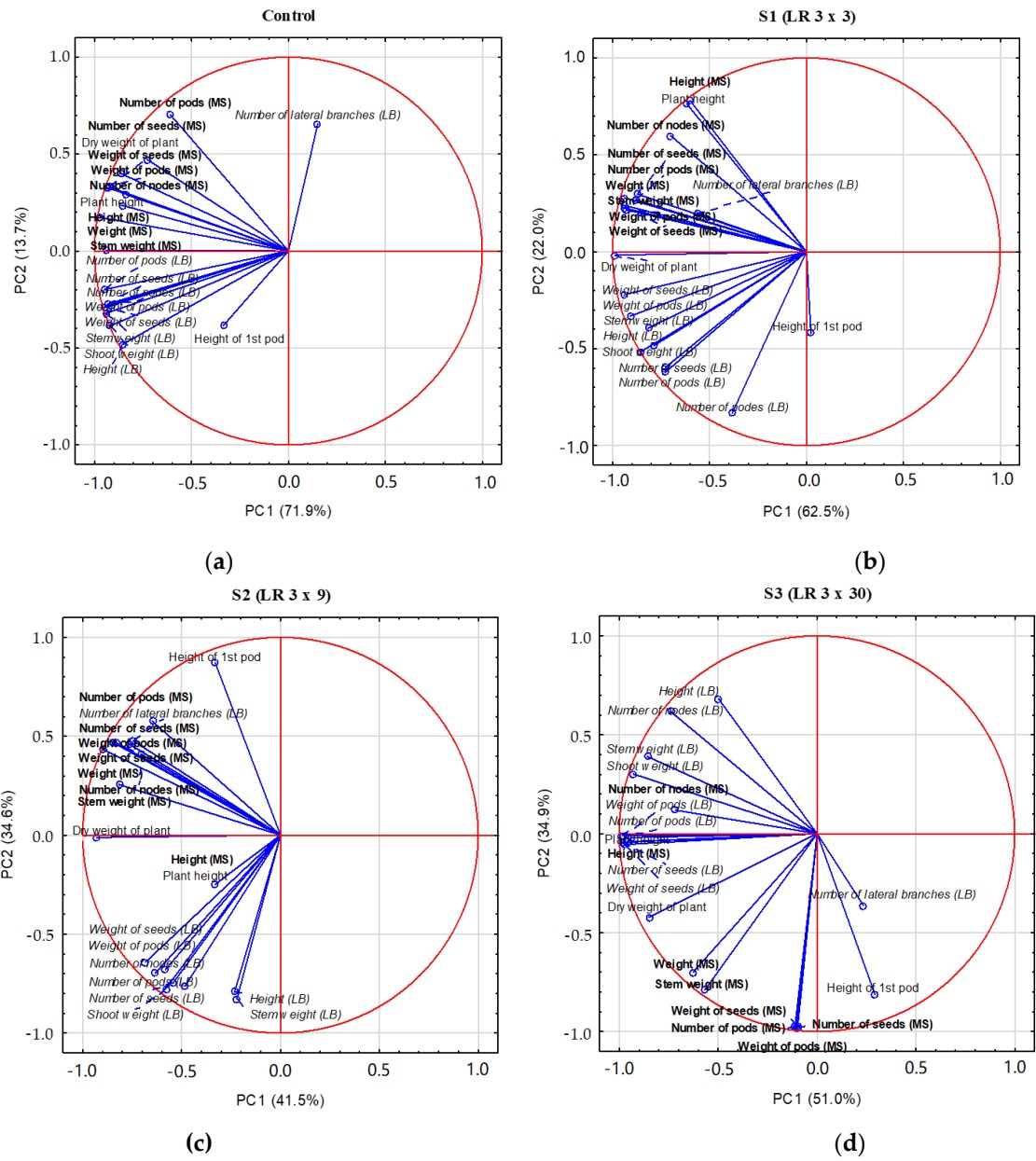

**Figure 3.** *Cont.*

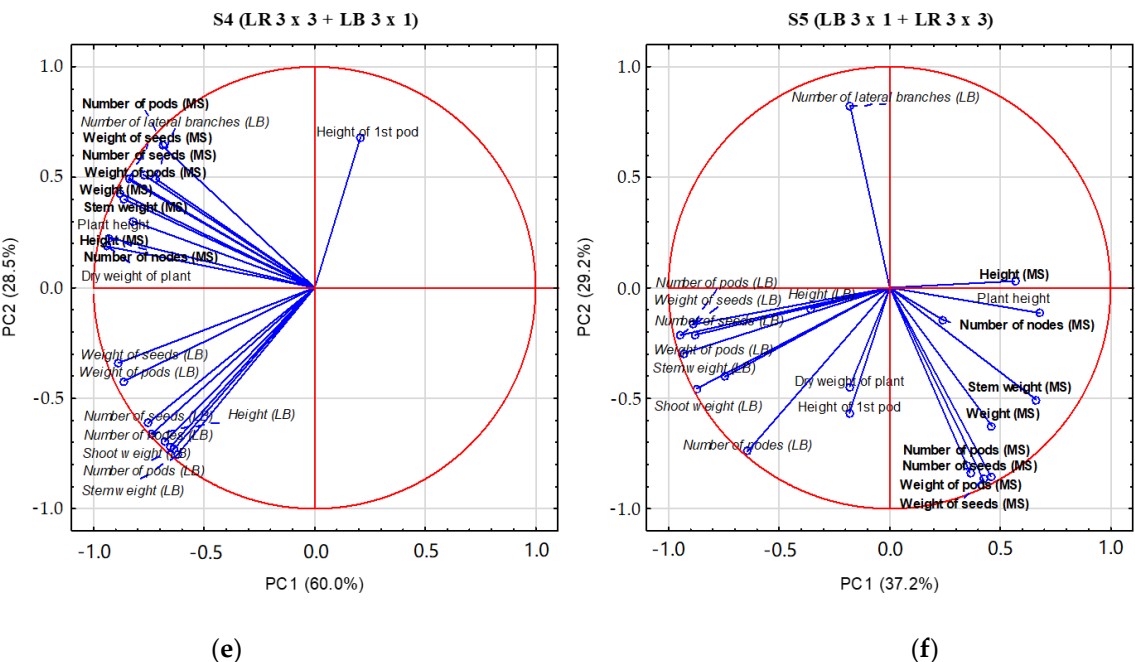

**Figure 3.** Principal component analysis of PC1 and PC2 on yield and biological parameters of soybean located on the main shoot (MS) and lateral branches (LBs), depending on laser stimulations: (**a**)— Control, (**b**)—S1 (LR 3 × 3), (**c**)—S2 (LR 3 × 9), (**d**)—S3 (LR 3 ×30), (**e**)—S4 (LR 3 × 3 + LD 3 × 1), (**f**)—S5 (LD 3 × 1 + LR 3 × 3).

In the first analyzed group (Control), PC1 explained 71.9% of variables. PC1 included the following variables for lateral branches which were negatively correlated with it: number of pods, number of seeds, number of nodes, weight of pods, weight of seeds, stem weight, shoot weight, height and included all biometric traits for main shoots expect the number of pod. The variable number of lateral branches was positively correlated with it. PC2 explained variables at 13.7%. PC2 included variables positively correlated with the number of pods on the main shoots.

In the second analyzed group (LR 3 × 3), PC1 explained 62.5% of the variables. PC1 included the following variables for lateral branches which were negatively correlated with it: dry weight of plant, weight of seeds, weight of pods, weight of stem, weight of lateral branches, number of seeds, number of pods, and included all biometric traits for main shoots expect the height. PC2 explained variables at 22.0% and was positively correlated with the height of main shoots and negatively correlated with number of lateral branches.

In the third analyzed group (LR 3 × 9), PC1 explained 41,5% of the variables. PC1 included the following variables for main shoots which were negatively correlated with it: number of seeds, weight of seeds, number of pods, number of nodes, and weight of plant. PC2 explained variables at 34.6%. PC2 was negatively correlated with variables for lateral branches: weight of lateral branches, height, number of nodes, number of pods, weight of pods, and number of seeds.

In the fourth analyzed group (LR 3 × 30), PC1 accounted for 51.0% of the variables. PC1 included the following variables for lateral branches which were negatively correlated with it: seed weight, weight of seeds, weight of pods, weight of branches, and height. PC2 explained variables at 34.9%. PC2 mainly included the following negatively correlated variables for the main shoots: weight of first pod, shoot weight, stem weight, and seed weight.

In the fifth analyzed group (LR 3 × 3 + LD 3 × 1), PC1 accounted for 60.0% of variables. PC1 included the following variables for lateral branches which were negatively correlated with it: seed weight, number of seeds, pod weight, and height, as well as variables for main shoots that were negatively correlated with it: seed weight, seed number, pod weight, pod

number, stem weight, and weight. PC2 explained variables at 28.5%. PC2 was negatively correlated with the following variables for lateral branches: number of pods and weight of branches.

In the sixth analyzed group (LD 3 × 1 + LR 3 × 3), PC1 explained 37.2% of the variables. PC1 included the following variables for the main shoots which were positively correlated with it: height, number of nodes, and pod weight; and the following variables for lateral branches that were negatively correlated with it: number of pods, pod weight, number of seeds, and seed weight. PC2 explained variables at 29.2%. The following variables for the main shoots were negatively correlated with it: pods weight, number of pods, seed weight, and number of seeds.

## 4. Discussion

Physical pre-sowing light stimulation of soybean seeds has a significant effect on plant morphology, seed yield, and yield quality. Light stimulation was shown to have greater effects in shaping the morphological parameters of the lateral branches of the plants and lesser effects in shaping the morphological parameters of the main shoots. An interesting result of the research was the discovery of a positive effect of pre-sowing seed stimulation on the diversification of plant morphological features, not only as a comprehensive effect, but also in terms of the differentiation of parameters on lateral branches. This is a contribution to the development of scientific research in this field. The most beneficial effects of stimulation on the development of plant weight were noted in the plants stimulated with combinations S3 (LR 3 × 30) and S5 (LD 3 × 1 + LR 3 × 3), which had direct impacts on yield structure and seed yield. The positive synergistic effect of using different laser light types (red and blue) in sequence was observed by Dobrowolski et al. [28]. The authors proved that the optimal combination of an algorithm used for the biostimulation of biological material significantly increased the effectiveness of bioremediation processes in aquatic ecosystems. The authors focused more on the effects of laser stimulation, without analyzing the biological mechanisms that are activated by different wavelengths of light. The photostimulation of biological materials with a blue diode with a wavelength of 437 nm was found to stimulate mitochondrial enzymes, i.e., cytochromes [7]. Blue light is highly absorbed by chlorophyll a and b and has an impact on photosynthetic rate [29]. A red laser with a wavelength of 660 nm improves DNA structure in plant cells, which has impacts on plant growth and development. Lanoue et al. [29] noticed that species react in various ways to light stimulation of different wavelengths, and more interdisciplinary studies are needed to explain the biochemical background of this phenomena. Śliwka [30] assessed the impact of coherent light on resistance in plants growing in unfavorable environmental conditions. The author noticed that the germination capacity of *Phelum pratense* showed a beneficial effect of the red laser diode (670 nm), while a shortened germination time was obtained as a result of irradiation with the blue laser diode (437 nm). These phenomena were not observed in our studies in the germination stages of the soybean plants, but relevant differences among treatments were noticed at the maturation stages.

The numbers and weights of pods, as well as the numbers and weights of seeds, were significantly higher in the S3 and S5 treatment groups than in the control. Similar results—though presented as main effects for the plants—were reported for the pre-sowing light stimulation of pea seeds using a He-Ne laser with respect to the biochemical processes of seedlings and the morphological traits of the plants [31]. Exposure of seeds to He-Ne laser light improved the rate and uniformity of germination and modified the stages of growth, which accelerated the flowering and maturation of the peas. Light stimulation with laser light improved certain morphological traits of the plants, such as plant height and leaf area, as well as overall yield. The increase in seed yield was due to increased numbers of pods and seeds per plant, with no significant changes noted in the numbers of seeds per pod [31]. Pre-sowing light stimulation of seeds has also been shown to increase yields in other crops. The seed yield of oats was increased through irradiation relative to

the control by 16.5% to 23.2% [32]. Dziamba et al. [33] showed a 7.9% to 25.5% increase in the grain yield of maize compared to the control. Studies on the efficiency of pre-sowing light stimulation of alfalfa seeds and red clover seeds also showed increases in the green and dry matter yields of the plants [33–37]. Podleśny [38] observed a seed yield increase in broad bean ranging from 5.1% to 9.4% in comparison with the control, depending on how many times the seeds were irradiated before sowing. A positive effect of light stimulation on broad bean yields was also confirmed in another study by Podleśny [24]. Our findings confirmed the aforementioned results, since significantly higher total seed yields were obtained with longer irradiation times (combination S3 (3 × 30)) for soybean seeds compared to the control.

The most important quality characteristics of soybean seeds are the contents of crude protein and crude fat. The values for these nutrients recorded in studies conducted in Central Europe are about 37.1% and 18.2–19.7%, respectively [39]. A comparison of these averages with the results of our study shows that the contents of crude protein and crude fat in the soybean seeds were slightly different from the averages. Light stimulation treatment S3 (LR 3 × 30) increased the contents of crude protein and fat in the seeds. In contrast, Dziamba and Dziamba [40] observed that irradiation of rapeseed had no significant effect on the quality characteristics of the seeds produced.

## 5. Conclusions

Light stimulation of seeds before sowing significantly influenced the morphological parameters, seed yield, and seed quality of soybean plants. The best effects on production were obtained using the combination LB 3 × 1 + LR 3 × 3, with which treatment yields were doubled compared to the control. A beneficial effect of combined (blue–red) light stimulation on plant morphology, which differentiated the plant productivity of the lateral branches, was revealed.

The combination with the most beneficial effect on crude protein content was LR 3 × 30, while the greatest increase in accumulated crude fat content in the seeds was obtained using the combination LR 3 × 9. Light stimulation of soybean seeds can play a significant role in determining plant morphology and have a relevant impact on seed yield, as stimulation was shown to result in the high plasticity of the lateral branches of the plants.

More studies are needed to explain the biochemical background of using sequential light stimulation, given the manifest effects of such approaches that have been reported. Our findings will be very useful for breeders working on new genotypes suitable for cultivation in environments with the use of new light technology.

**Author Contributions:** Conceptualization, A.K.-K. and J.W.D.; investigation, A.K.-K. and A.Ś.; formal analysis, A.K.-K., D.K., A.Ś. and Z.P.; data curation, A.K.-K.; writing—original draft preparation, A.K.-K. and R.W.N.; methodology, A.K.-K.; visualization, A.K.-K.; supervision and project administration, A.K.-K.; writing—review and editing, A.K.-K., R.W.N. and A.Ś.; statistical analysis, E.D. and A.K.-K. All authors have read and agreed to the published version of the manuscript.

**Funding:** The research was funded by the Ministry of Science and Education, Poland.

**Institutional Review Board Statement:** Not applicable.

**Informed Consent Statement:** Not applicable.

**Data Availability Statement:** Not applicable.

**Conflicts of Interest:** The authors declare no conflict of interest.

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
