# Peer review of "Pre-Sowing Laser Light Stimulation Increases Yield and Protein and Crude Fat Contents in Soybean"

_agriculture, doi:10.3390/agriculture12101510_

Round 1

Reviewer 1 Report (Previous Reviewer 2)

This is a much improved version of the manuscript than the original that I recently reviewed. Authors seem to have addressed all of my comments adequately, resulting in a good manuscript.

Author Response

Dear Reviewer,

Thank you very much for suggestions and comments. Some extra changes were made accoridng to other Reviewer suggestion.

Kind Regards, Authors

Reviewer 2 Report (New Reviewer)

As the direct treatment of seed is linked with emergence/germination attributes. In this experimentation, data of emergence attributes are not presented; a major drawback of experiment. If data are available, may be added in the revised version accordingly. For further improvements, consider the following comments and suggestions.

Abstract

Line 22: “three times (3×3 sec)” is not clear. What the authors want to say?

Line 29: three times (3×9 sec)?

Problem statement is missing. Add a brief description about the importance/role of combinations of light stimulation in the start of abstract.

Introduction

Page 2 Line 4-5: incomplete sentence.

Line 10: improving seeds, not clear.

Importance/chemical composition of soybean seeds may be added in the introduction section as quality parameters are observed during the experimentation. The introduction may be improved with scientific approach to ensure the quality of draft.

Materials and Methods

Line 42: 3 second.

Page 3 Line 3-4: follow same pattern.

Line 13-14: add space between values/digits and units.

Procedures of biometric parameters and traits of shoots are not explained. Details (instruments etc.) may be added. Briefly explain the protocols used for the estimation of crude protein and crude fat contents with reference.

Results

Table 2: Units are not clear. What do you mean by total yield? Is it biological yield, grain yield or their sum? Make it clear. Also add the comparison value, LSD, HSD etc.?

Data presented in Table 2 and 6 regarding seed weight and 1000 seed weight in not supporting each other, cross match please.

It is better to interpret the significant findings rather than all results.

Discussion

Line 11-14: The mentioned parameters are not studied in the current experiment. Add reference.

This section looks too poor as no mechanism either physiological and genetic is described. It looks just the findings of other studies. Just results interpretation is there.

Conclusion

Conclusion is not properly presented. It includes the recommendation and suggestion. It may be quantified. Revise the conclusion accordingly.

Author Response

Response to Rev.2

We would like to thank for comments. We corrected manuscript according to Reviewer suggestion.

Abstract

The brief description about importance of combinations of light stimulation was inserted.

The lines 22 and 29 are cleaned from not clear information.

Introduction 

The incomplete  sentence in line 4-5 P. 2 was corrected.

Information about ‘improving seeds’ was expended.

The importance of chemical staudies of soybean was added.

Materials and Methods

The information about emergency was added.

‘The irradiated seeds emerged at the same time - two weeks after sowing, while the seeds from the control treatment emerged five days later.’

Time unit is presented in SI unit (s), second. It is not nesesery to  changed it.

The procedure of biometric parameters was made handy. The following biometric parameters were determined manually using lab scales and digital ruler.

Results

Total yield is a sum of main and lateral shoots/branches. We use Tuke’s test and in that case another statistic is not necessary, since according to statistical methodology statistical tests are using if the treatments differed significantly

In Table 2 and Table 6 the results are correctly presented.

Discussion

The sentence from Line 11-14 was deleted.

The discussion can  not include the physiological or genetic description, since the aim of the study was dedeciated to another issues.

Conclusion

Conclusion was corrected. Some information was added.

Reviewer 3 Report (New Reviewer)

Laser biostimulation technique can be used to increase the adaptability and cultivation of crops under sub-optimal conditions. Physical parameters of laser (wavelength, power density, output power, beam type, irradiated area, fluence) are equally important to determine the effect of laser biotechnology.

In the introduction on Page 1, line 40; the authors have mentioned specific temperature and moisture requirements for soybean cultivation. Authors should mention optimal climatic conditions required for soybean cultivation and compare them with sub-optimal conditions in the study area (Poland). This will make a strong basis for use of laser irradiation as a pre-sowing treatment.

In methodology, how seed treatment was done in lab conditions? Please explain or add a reference in which the standardized method is mentioned. Though it is a field study, still the plot size is very small (7m2) and only 10 plants are selected for data recording. Ten plants may be good for recording biometric observation. However, for yield estimation, authors should record yield data of net plat size. The 30% jump in seed yield seems to be too much. It means varietal improvement is nothing in front of this (laser biostimulation) technology.

In the results chapter, the underlying reasons for such a response are not given. Why there is such stimulation in the present study by different treatment combinations? The observation of increased temperature or biological activity or changes at the cellular level can be there to validate the results.

LB and LD are used for the same treatment of Argon laser (blue laser light). use one. Further, S4 and S5 treatment is different for a sequence of laser types. In S4, LR is followed by LD while LD is followed by LR in S5. Results indicated that S5 is better than S4. Why is it so? Any comments?

The weather parameters are showing variability over two years. Authors should discuss the reasons for the non-significant effect of weather variability on soybean growth and yield.

Table 8 is not there. kindly add it.

In the discussion chapter, the authors have mentioned basic cell processes responsible for light stimulation on Page 9, lines 11-14. There is no supporting data for increased cell division of irradiated seeds or faster uptake of nutrients.

In the Conclusions on Page 10, lines 42-44, the authors have mentioned again about sub-optimal conditions. Kindly elaborate.

Author Response

Response to Rev.3

We would like to thank for comments and suggestion. We included suggestions in manuscript.

In the introduction section we modyfied the sentence about ‘sub-optimal’ conditions. We added explenation about reason of undertaken such studies.

In methodology section we added reference dedicated to seed treatment in lab conditions. We also corrected information, which were previously omited.

LB and LD are not the same. LB- lateral branches, while LD laser diode of blue light

The table 8 was replaced by Fig. 2. We deleted table 8 from text.

The added en explenation of reason, why we omited in result section  data presented in each year of study. 

The results of the research were presented as average over the years due to the lack of unequivocal differentiation of the weather conditions in the prevailing growing seasons. No significant interaction between treatments and weather was detected. Thus, the results were presented as averages.

In discussion section We also add reference and highlighted that is missing explenation in literature, why the sequention (red- blue or blue-red) pre-sowing laser stimulation has impact on plant features. 

The conclusion was corrected. We deleted term ‘sub-optimal’, since we did not analysis sub-optimal environmental conditions of soybean cultivation.

Round 2

Reviewer 2 Report (New Reviewer)

Most of the comments are addressed by the authors.

This manuscript is a resubmission of an earlier submission. The following is a list of the peer review reports and author responses from that submission.

Round 1

Reviewer 1 Report

The authors conducted experiments to evaluate the effects of laser biostimulation on soybean seeds in increasing soybean yield and improving soybean quality. This paper lacks sufficient justification for the research's importance and experimental designs. Critical experimental and analytical details were missing all the way through the manuscript. All the factors raised questions about the scientific soundness of the results and discussion. Detailed can be found in the attachment. 

Author Response

Dear Reviewer,

We would like to thanks for comments and suggestions. 

The corretion of manuscript was done. We added sufficient justification for the research's importance and experimental designs.

Critical experimental and analytical details were also added all the way through the manuscript.

We hope that the scientific soundness of the results and discussion is visible. In the manuscript all changes were marked.

Kind regards,

Authors

Reviewer 2 Report

The manuscript "agriculture-1799735" authored by  Klimek-Kopyra et al. reports on a study in using biostimulants to affect yield and seed compositional traits in soybean. It is an interesting manuscript that reports significant results, which may be of interest to the soybean research community. For the most part, the manuscript is well written. The best sections are introduction, results and discussion and the data are presented clearly. However, I have issues with the poor description of experimental design used, the size and area of field plots, the field layout, sampling procedure and the choice of the cultivar used. I have included numerous comments in the attached PDF for authors' consideration in order to improve the manuscript. They should be successful in addressing the comments.

Author Response

Dear Reviewer,

We would like to thanks for comments and suggestions. 

The correction of manuscript was done. We added sufficient justification for the research's importance, experimental designs as well methodology section.

Critical experimental and analytical details were also added all the way through the manuscript.

We hope that the scientific soundness of the results and discussion is visible. In manuscript all changes were marked.

Kind regards,

Authors

Reviewer 3 Report

In the manuscript titled Pre-sowing laser biostimulation increases the yield, protein, and fat content of soybean the authors describe the process of biostimulation of soybean seeds using a laser to increase the yield and content of protein and fat in soybean grain. They use two types of lasers, two ways of irradiating seeds, and different irradiation times. Based on the presented results, they conclude that the effect of both lasers is the realization of about 30% higher soybean yield as a result of the increased number of shoots on stems, increased number of pods on them, and an increased number of grains. Moreover, they claim that the yield is higher on the main shoots than on the side ones. They also conclude that laser biostimulation increased the protein content by 11% and fat by 5% in soybeans.

However, the manuscript suffers from quite essential and methodological shortcomings. I will list a few I consider essential that the authors need to elaborate on further, clarify, and specify.

1. The authors do not state anything about the model used i.e. how many soybean grains were laser treated with different procedures including the grains number of the control groups.

2. The authors do not specify germination efficiency in all groups (the treated and control) and how many seeds from all groups were finally planted and later stems entered the evaluation process.

3. Related to both previous objections are the results presented in all tables. Namely, the parameters followed and measured were not given with belonging variability (the standard deviation, SD). This makes it impossible to gain insight statistically into whether the observed change is a consequence of laser biostimulation or is within natural variability. The results are shown with one or two decimal places but SD is not shown.  Here are several examples. Table 3, column 2, the height of the first pod: average height of laser-treated is 6.8 with an SD 1.3 (the control is 6.57). Table 5, the shoot weight, average 10.5, SD 5.5, control is 4.6. Table 7, the pod number, average 2.1, SD 0.3, control 1.94. And so on.  The authors state that the values given in the tables are means but not how many samples are included. Significantly different means (P less than 0.05) are marked by different letters, a, b, c, d, ab, bc, abc, but their meanings are not explained.

4. What are the beam diameters of the lasers that are used to treat the seeds? Based on which is the given treatment model was selected?

5. For the chemical composition, you state the protein and fat content in the seeds. What proteins and fats? The composition of the seed of each plant is very complex and includes many proteins and fats, but you do not give any procedures for processing the seeds to extract the specific protein or proteins, or fats that you want to follow. Give more detail and a precise description of the procedure and instrumental methods that were used. Related to that, Table 8 does not show a statistical difference between the treated and control groups. For example, the protein content of main-to-lateral shoots is 356/330, SD 25, and control is 333/318; the average protein-to-fat from the plant is 343/176, SD 26/4, control 325/164.

6. In The folder Results, the authors discuss various positive effects of biostimulation expressed in one decimal place percentages. Given the above remarks, I consider it impossible to draw such precise conclusions. 

7. In the Discussion, the authors mostly cite the results of other authors without significantly elaborating on their results.

8. In Conclusion, the authors emphasize a significant increase in total soybean yield when the grain is treated with an appropriate combination of lasers (S5). However, a similar increase was registered with S2, Table 2, but the authors do not state that in the conclusion. Related to this is the general conclusion that due to the non-presentation of the working model, i.e. the number of seeds treated, involved in the experiment it is not possible to extract accurate and precise conclusions about the effects of laser-based biostimulation of the soybeans. The results are very dispersive and I suggest to authors apply PCA analysis in order to extract accurate enough and relevant conclusions.

Author Response

Dear Reviewer,

We would like to thanks for comments and suggestions. 

The correction of manuscript was done. We added sufficient justification for the research's importance and experimental designs. We also added PCA analysis according to Your suggestion .

Critical experimental and analytical details were also added all the way through the manuscript.

We hope that the scientific soundness of the results and discussion is visible. In manuscript all changes were marked.

Kind regards,

Authors

Reviewer 4 Report

See comments on paper.

Title: Seed pre-sowing light stimulation increases yield, protein and fat content of soybean.

1. Is the subject addressed in the article relevant to be published by the journal?
Yes, the article shows a new idea.

2. Is the article original?
Yes, the article open the new opportunities to crop plants

3. Does the title clearly and sufficiently reflect the content of the article?
No. the term ”biostimulation” would be changed for “light stimulation” because that the source of stimuli are physic not bio! Revise all article.

4. Is the presentation, organization and size of the article satisfactory?
Yes, all parts are proportional and sufficient.

5. Does the introduction review the topic addressed and make clear the purpose of the work?
Yes, but the term “aerial part” may be changed for “shoot” – revise all article

6. Is the item material and methods sufficiently clear to allow the article to be reproduced?
Used seed to sow the experiment. But what it harvested and evaluated is grain. So how much to refer to the harvested product should write as grain. Revise all article.
Are a mistake, because at beginning of results highlighted of season are particularly different and the data’s of soybean obtained from each year are presented as average. This masks/hides important information. Leaving poor the interpretation and discussion of the results and consequent discussion.
Revise term “lateral shoots”; change for side branches or lateral branches;
Revise column data on table 3 and table 4 as plant height - Seems to me to repeat information, except for treatment S5; even if it is correct or this data was obtained from different plants - the information is duplicated. So, just one of them is enough.
Review the writing of the results, because in some moments of writing it addresses values in percentage, as in the case of table 1 yield on main shoot and TGW as 102% and 94%, but by Tukey does not differ.

7. Is the discussion relevant and sufficient?
Yes, but depend on if data’s are opened to probable interaction from each season.

8. Do the data justify the interpretations?
Yes, but you can explore in depth.

9. Is there any need to add something that can enrich the article?
Yea; see previous comments.

10. Is it necessary to reduce or remove any part of the article?
No.

11. Are illustrations and tables necessary and relevant?
See previous comments.

12. Are the figures illustrative and are they of good quality for reproduction?
Not applicable- because only tables.

13. Are the keywords suitable for the article?
Revise- if possible not repeat. Do not repeat those in the title. Example biostimulation. By luminous stimulation.

14. Does the abstract give good information about the work?
Yes.

15. Are references adequate and necessary?
See previous comments – about the seasons effects.

16. Are the references written in accordance with the journal's rules?
Yes, about my knows.

17. Are the authors referenced in the text cited in the references?
Yes.

18. Was any annotation made in the manuscript?
No.

Author Response

(The authors gave the same response as above.)
